# Is babesiosis a rare zoonosis in Spain? Its impact on the Spanish Health System over 23 years

Hugo Almeida[1☯], Amparo López-Bernús[1☯], Beatriz Rodríguez-Alonso[1], Montserrat Alonso-Sardón[2], Ángela Romero-Alegría[1], Virginia Velasco-Tirado[3], Javier Pardo-Lledías[4], Antonio Muro[5], Moncef Belhassen-García[1]*

1 Servicio de Medicina Interna, Hospital Universitario de Salamanca, Salamanca, Spain, 2 Área de Medicina Preventiva y Salud Pública, Universidad de Salamanca, Salamanca, Spain, 3 Servicio de Dermatología, Hospital Universitario de Salamanca, Salamanca, Spain, 4 Servicio de Medicina Interna, Hospital Marques de Valdecilla, Universidad de Cantabria, IDIVAL (Instituto de Investigación Valdecilla), Santander, Spain, 5 Grupo de Enfermedades Infecciosas y Tropicales (e-INTRO), Instituto de Investigación Biomédica de Salamanca (IBSAL), Salamanca, Spain

☯ These authors contributed equally to this work.
* belhassen@usal.es

**Data Availability Statement:** All relevant data are within the paper and its Supporting Information file.

## Abstract

### Background

Babesiosis is a zoonosis caused by an intraerythrocytic protozoan of the genus *Babesia* and transmitted mainly by ticks of the *Ixodes* spp. complex. There is no comprehensive global incidence in the literature, although the United States, Europe and Asia are considered to be endemic areas. In Europe, the percentage of ticks infected with *Babesia* spp. ranges from 0.78% to 51.78%. The incidence of babesiosis in hospitalized patients in Spain is 2.35 cases per 10,000,000 inhabitants/year. The mortality rate is estimated to be approximately 9% in hospitalized patients but can reach 20% if the disease is transmitted by transfusion.

### Objective

To analyze the epidemiological impact of inpatients diagnosed with babesiosis on the National Health System (NHS) of Spain between 1997 and 2019.

### Methodology

A retrospective longitudinal descriptive study that included inpatients diagnosed with babesiosis [ICD-9-CM code 088.82, ICD-10 code B60.0, cases ap2016-2019] in public Spanish NHS hospitals between 1 January 1997 and 31 December 2019 was developed. Data were obtained from the minimum basic dataset (CMBD in Spanish), which was provided by the *Ministerio de Sanidad*, *Servicios Sociales e Igualdad* after the receipt of a duly substantiated request and the signing of a confidentiality agreement.

**Funding:** The authors received no specific funding for this work.

**Competing interests:** The authors have declared that no competing interests exist.

## Main findings

Twenty-nine inpatients diagnosed with babesiosis were identified in Spain between 1997 and 2019 (IR: 0.28 cases/10,000,000 person-years). A total of 82.8% of the cases were men from urban areas who were approximately 46 years old. The rate of primary diagnoses was 55.2% and the number of readmissions was 79.3%. The mean hospital stay was 20.3 ±19.2 days, with an estimated cost of €186,925.66. Two patients, both with secondary diagnoses of babesiosis, died in our study.

## Conclusions

Human babesiosis is still a rare zoonosis in Spain, with an incidence rate that has been increasing over the years. Most cases occurred in middle-aged men from urban areas between summer and autumn. The Castilla-La-Mancha and Extremadura regions recorded the highest number of cases. Given the low rate of primary diagnoses (55.2%) and the high number of readmissions (79.3%), a low clinical suspicion is likely. There was a 6.9% mortality in our study. Both patients who died were patients with secondary diagnoses of the disease.

## Introduction

Babesiosis is a zoonosis caused by an intraerythrocytic protozoan of the genus *Babesia, which is manly* transmitted by ticks of the *Ixodes* spp. complex [1, 2]. The distribution of *Ixodes* ticks, the most prevalent on the world, has been on the rise since the beginning of the 20th century, and *Ixodes ricinus* is the most prevalent tick species worldwide [3, 4].

In recent European studies, *B. divergens*, *B. microti* and *B. venatorum* have been identified as the most prevalent agents [1, 5–7]. The main hosts were cattle roe deer and small mammals, like mice. In Europe, the percentage of ticks infected with *Babesia* spp. ranges from 0.78% in Switzerland to 51.78% in Austria [7–17].

There are four primary mechanisms of transmission to humans: i) tick bite (the most common); ii) blood transfusion or iii) solid organ transplant (both from an infected donor, asymptomatic carriers); and iv) transplacental (rare) [18]. Up to 9% of *B. microti* infections that require hospital admission are fatal, increasing to 20% in cases of transmission through transfusions [19]. The mortality rate due to *B. divergens* infection increases in splenectomized patients. The disease can be diagnosed in the acute phase by peripheral blood smear or polymerase chain reaction [20] and in the convalescent phase by seroconversion [20–22]. Treatment is based on the combination of an antiparasitic (atovaquone) and an antibiotic (azithromycin) [23]. Oral quinine and intravenous clindamycin may be alternatives when there is clinical evidence of failure to respond to atovaquone [24].There is no prophylactic antibiotic regimen or vaccine approved thus far [25].

The worldwide incidence of human babesiosis is unknown. In the United States, where it is a notifiable disease, a total of 14,042 cases were reported between 2011 and 2017 [26]. In Europe, in a seroprevalence screening carried out in Belgium in 2014, the results were 9% (*B. microti*), 33.2% (*B. divergens*) and 39.7% (*Babesia* spp. EU1) [27]. In Spain, the incidence reported thus far is 2.35 cases/10,000,000 inhabitants/year in hospitalized patients [28], although this figure is believed to be underestimated. Since 2003, only 8 diagnosed cases of babesiosis have been published in Spain, including 3 cases of *B. divergens* and 5 cases of *B. microti* [7, 29–35]. The risk factors studied include certain high-risk professions, such as farming, ranching and veterinary

medicine, in addition to individuals who reside in or travel to endemic areas [36, 37]. Further-more, most tick borne babesiosis occurs between the months of May and September [38, 39].

The aim of this study was to evaluate the epidemiological status of babesiosis in patients in Spain between 1997 and 2019.

## Material and methods

### Study design and population

This is a retrospective longitudinal descriptive study of hospitalized patients diagnosed with babesiosis in public hospitals of the Spanish National Health System (NHS) between January 1, 1997, and December 31, 2019.

**Inclusion criteria:** All patients admitted to NHS public hospitals between 1997 and 2019 with a principal and/or secondary diagnosis of babesiosis according to the *International Classification of Disease* codes (9th edition, Clinical Modification (ICD-9-CM), code 088.82, for cases from 1997–2015, and 10th edition (ICD-10), code B60.0, for cases from 2016–2019). **Exclusion criteria:** Patients with missing data were excluded from the study.

### Data collection

This study analyzes the data provided by hospital discharge records (HDR). HDR includes all hospital discharges from the NHS network of general hospitals. Data were obtained from the Minimum Basic DataSet (CMBD in Spanish). CMBD is the main database for information on morbidity data for patients treated at these hospitals, and on the care process for these patients. It provides usual demographic data (age, sex, and place of residence), clinical variables (diagnoses and procedures) and variables related to the episode of hospitalization, such as nature of the admission (urgent or scheduled), patient discharge (discharge to patient's residence, transfer to another hospital, or death), and average length of stay. The diagnoses and procedures collected were coded using the *International Classification of Diseases*, 9th *Revision*, *Clinical Modification* (ICD-9-CM) and 10th *edition* (ICD-10). *Principal diagnosis* was defined as the condition after the study, which led to hospital admission, according to the ICD-9-CM/ICD-10 Official Guidelines for Coding and Reporting. [It is important not to confuse the principal diagnosis with the primary diagnosis. *Primary diagnosis* is the most serious and/or resource-intensive during hospitalization or inpatient consultations]. *Secondary diagnoses* (up to 13) are *"other diagnoses"* or conditions that coexist at the time of admission or develop after admission and that affect patient care during the current episode.

### Data analysis

The *incidence rate* was calculated by dividing the number of new cases of babesiosis per year/period (numerator) by the population at risk for the disease within the same given period (denominator; person-years) multiplied by 10,000,000 and expressed as "cases per 10,000,000 person-years". As it is not possible to accurately measure disease-free periods, the total person-time at risk can be estimated approximately and satisfactorily when the size of the population is stable by multiplying the average population size studied by the duration of the observation period. Thus, the population at risk was obtained from annual data published by the National Institute of Statistics (INE, http://www.ine.es/). ☙The estimated average population of Spain for the 1997–2019 period was 44,545,650 inhabitants; 21,910,723 men and 22,364,926 women]. The 95% confidence interval (95% CI) for the incidence rate was calculated for a better clinical application of the results. Incidence rates were computed according to autonomous community and year to assess the temporal and geographical patterns. The results in terms of

mean incidence rates by autonomous community were plotted in maps for the entire study period.

The *lethality rate* was calculated by dividing the number of deaths according to principal diagnosis (numerator) by the amount of affected individuals with a principal diagnosis of a specific disease (denominator) (x100).

The results were expressed as absolute value (n) and percentage (%) for categorical variables and as the mean, standard deviation (SD), median, interquartile range (IQR) ($Q_3$-$Q_1$), range (minimum value, maximum value) for continuous variables. A chi-square ($\chi^2$) test was used to compare the association between categorical variables, such as clinical and demographic variables, and the measured outcome was expressed as the odds ratio (OR) together with the 95% CI for OR. Continuous variables were compared with Student's t test or the Mann–Whitney test for two groups, depending on their normal or nonnormal distribution. ANOVA allowed us to analyze the influence of nominal independent variables on a continuous dependent variable. Additionally, we applied the corresponding logistic regression model for multivariate analyses of categorical variables. We considered a difference to be statistically different from chance at a p value <0.05. Data analysis was performed using SPSS 26 (Statistical Package for the Social Sciences).

### Ethics statement

This study is based on the medical data of patients collected in the CMBD. These data are the responsibility of the Ministry of Social Services, Health and Equality (Ministerio de Servicios Sociales, Sanidad e Igualdad, MSSSI), which stores and organizes them. All patient data provided by the CMBD are anonymized by the MSSSI before they are provided to the applicants. According to this confidentiality commitment signed with the MSSSI, researchers cannot provide the data to other researchers, so other researchers must request the data directly from the MSSSI. The protocol and ethics statement of this study were approved by the Clinical Research Ethics Committee of the Complejo Asistencial Universitario de Salamanca (CAUSA). Because the data were obtained from an epidemiological database, written consent was not obtained. All data analyzed were anonymized.

## Results

### Temporal, seasonal, and geographical distribution

A total of 29 cases with ICD-9-CM code 088.82 and ICD-10 code B60.0 were registered in Spain during the 23-year study period, 1997–2019. The incidence rate for the period was 0.28 (95% CI, 0.18–0.38) cases per 10,000,000 person-years. The highest annual incidence rates were 1.07 (95% CI, 0.13–2.01) cases per 10,000,000 person-years (5 cases) in 2015 and 0.99 (95% CI, 0.02–1.95) cases per 10,000,000 person-years (4 cases) in 2000 (**Fig 1**). Babesiosis has a seasonal component, with a higher number of cases (72%) in the summer and autumn months (from June to November), although there are cases throughout the year. A total of 21 cases (72%) were reported in those months, 10% in the spring and 18% in the winter, during the study period (1997–2019).

**Fig 2** shows the geographic distribution of these 29 cases in Spain. The highest incidence rates were observed on the northern coast of the Iberian Peninsula (in Cantabria), with 1.53 (95% CI, -0.59–3.66) cases per 10,000,000 person-years, and in the western region (in Extremadura), with 1.20 (95% CI, -0.16–2.56) cases per 10,000,000 person-years. Up to 8 cases were reported in the province of Cordoba (southwest) and the incidence rate for the period was 4.45 (95% CI, 1.36–7.53) cases per 10,000,000 person-years (Autonomous Community of Andalucía). Four of these cases occurred in the same year (2000; 2 in January, 1 in April and 1 in July),

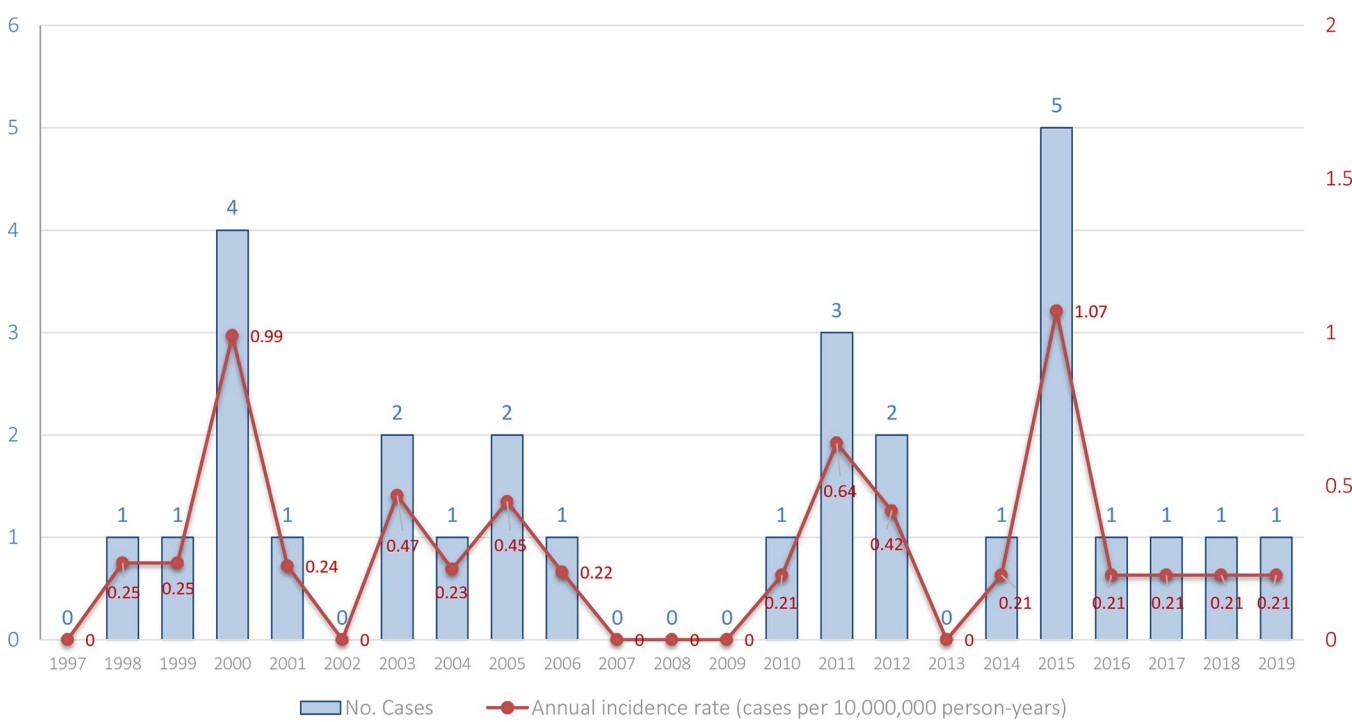

**Fig 1. Temporal distribution of human babesiosis in Spain, 1997–2019: Cases and annual incidence rate (cases per 10,000,000 person-years).**

and the other 4 cases occurred in different years (July 1998, June 1999, February 2003 and June 2005). Most of the cases (24 cases) were diagnosed in individuals from urban areas (population with more than 5,000 inhabitants), and only 1 case came from a rural area (population with less than 5,000 inhabitants); for 4 cases, the geographic origin was not recorded (**Table 1**).

## Distribution by age and sex

The number of cases was five times higher in men (82.8%) than in women (17.2%), with a male/female ratio of 4.8:1 (24/5) (**Table 1**). The incidence rate for men was 0.48 (95% CI, 0.28–0.67) cases per 10,000,000 person-years, whereas the incidence rate for women was 0.10 (95% CI, 0.01–0.18) cases per 10,000,000 person-years. The mean (±SD) age was 45.6 (± 17.6) years [median (IQR), 46 (62–28)], and the range was 17 to 86. Most of the cases (24; 82.7%) were adult patients (15–64 years old), and 5 (17.2%) were elderly patients (>65 years old). There were no significant differences in the mean age of men and women (45.6±17.1 vs. 45.8±21.7; p = 0.981).

## Distribution by diagnosis causing hospitalization

Hospitalizations due to a principal diagnosis of babesiosis (ICD-9-CM code 088.82 and ICD-10 code B60.0) included 16 (55.2%) cases, whereas babesiosis was a secondary diagnosis for 13 (44.8%) cases of hospitalization. The mean age of patients with principal diagnosis codes was lower than those with secondary diagnosis codes [mean±SD, 40.7±17.3 vs. 51.6±16.6; p = 0.098]. The average hospital stay was similar between patients with babesiosis as a principal diagnosis and those with babesiosis as a secondary diagnosis [mean±SD, 19.7±17.2 vs. 21.0 ±22.2; p = 0.859]. No significant differences (p>0.05) were observed in the independent qualitative and quantitative variables analyzed between the two groups of patients: principal diagnosis vs. secondary diagnosis (**Table 1**).

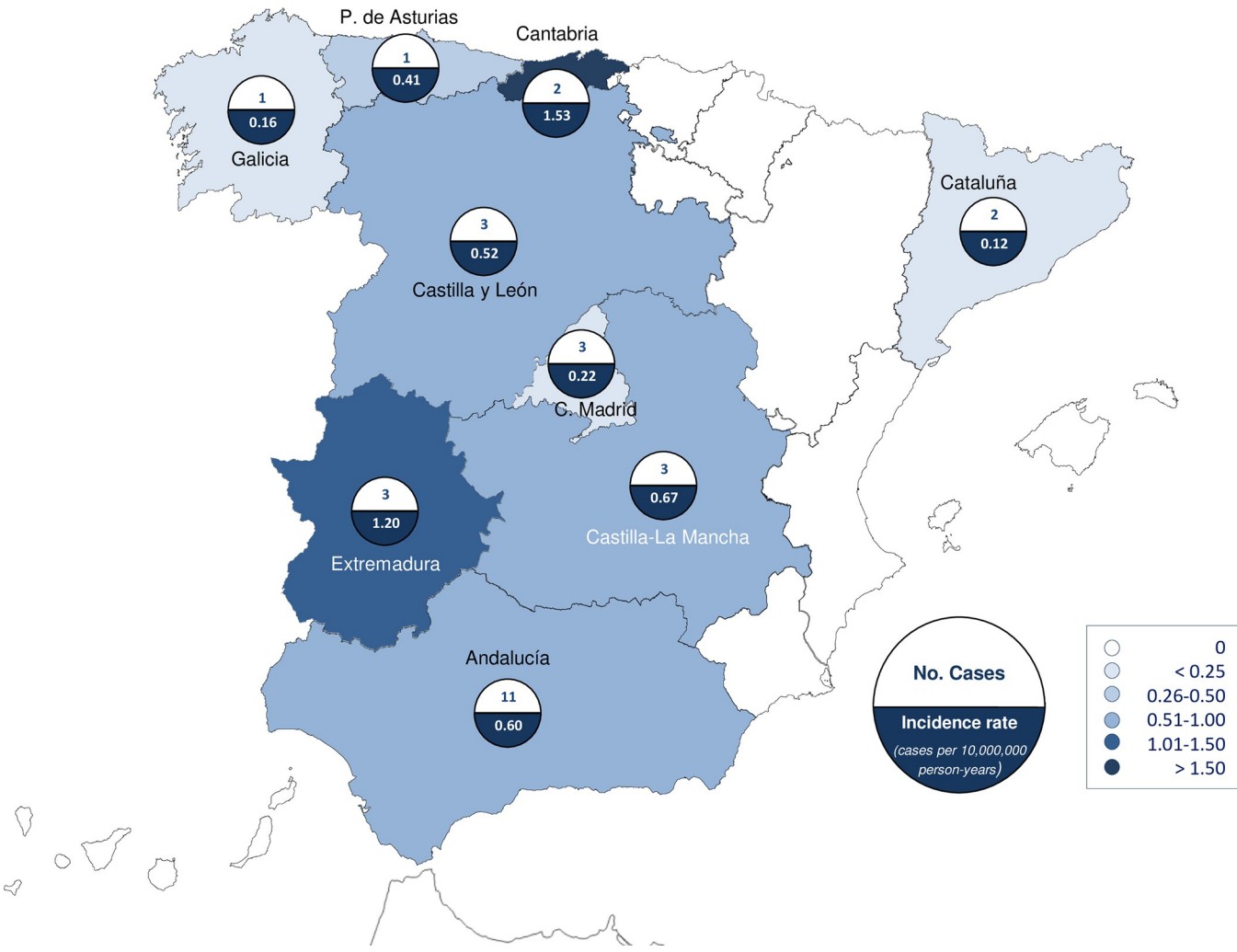

**Fig 2. Human babesiosis: No. cases and Incidence rates (cases per 10,000,000 person-years) by regions, Spain (1997–2019).**

### Clinical data

Regarding comorbidity, 3 (10.3%) patients had Lyme disease, and 4 (13.8%) cases had HIV coinfection. In relation to the nature of the hospital admission, 19 (65.5%) cases were urgent admissions, and 23 (79.3%) cases were hospital readmissions (30 days after a previous discharge). For 10 (34.5%) cases, there was no record of the service that discharged the patient. Of the 19 cases for which we had a record of the service treating the patient, 14 cases were under the care of the internal medicine service. Most cases (27, 93.1%) were non-surgical, 2 (6.9%) were surgical, and 25 (86.2%) were sent home after hospital discharge. The average (±SD) hospital stay was 20.3 (±19.2) days [median (IQR), 11 (31–8)] (**Table 1**).

### Cohort mortality

The overall lethality rate for the cohort was 6.9 per 100 (2 deaths/29 total cases of babesiosis). Regarding the two deaths, babesiosis was not the principal diagnosis at the beginning in either case. One patient had a clinical history of acute posthemorrhagic anemia and renal failure, and the other patient had cardiomyopathy.

**Table 1. Main data patients included in the study.**

| Variables | N = 29 cases (100%) |
|---|---|
| **AGE** | |
| Mean ± sd | 45.6 ± 17.6 |
| ≤17 years | 1 (3.4) |
| 18–64 years | 23 (79.3) |
| ≥65 years | 5 (17.2) |
| **GENDER** | |
| Male | 24 (82.8%) |
| Female | 5 (17.2%) |
| **RURAL VS. URBAN** | |
| Urban (< 5,000 inhabitants) | 24 (82.8%) |
| Rural (≥5,000 inhabitants) | 1 (3.4%) |
| Unknown | 3 (10.3%) |
| Foreigners | 1 (3.4%) |
| **CO-MORBIDITY** | |
| Human immunodeficiency virus (hiv) infection (042) | 4 (13.8%) |
| Lyme disease (088.81) | 3 (10.3%) |
| **TYPE OF HOSPITAL ADMISSION** | |
| Urgent | 19 (65.5%) |
| Programmed | 10 (34.5%) |
| **HOSPITAL SERVICE** | |
| Internal medicine | 14 (48.3%) |
| Infectious diseases | 2 (6.9%) |
| Intensive care medicine | 1 (3.4%) |
| Other services | 2 (6.9%) |
| Unknown | 10 (37.0) |
| **TYPE OF DISCHARGE** | |
| Home | 24 (82.8%) |
| Transfer to another hospital | 3 (10.3%) |
| **MORTALITY** | |
| Overall mortality | 2/29 (6.9%) |
| Babesiosis principal diagnosis mortality | 0/15 (0%) |
| **HOSPITAL STAY (DAYS)** | |
| Mean ± sd | 20.3 ± 19.2 |
| Median (iqr) | 11 (31–8) |
| Range (minimum value, maximum value) | (2, 82) |
| **COST (€)** | |
| Total sum | 186,925.66 |
| Mean ± sd | 6,445.71 ± 8,154.50 |
| Range (minimum value, maximum value) | (2,433.08, 36,955.56) |

## Cohort cost

We estimated the overall cost incurred by this cohort of patients (**Table 1**). In Spain, hospital-admitted patients with a diagnosis of babesiosis (from 1997 to 2019) had a total cost of approximately €186,925.66. The average (±SD) cost per patient was €6,445.71 (±8,154.50). The average cost was higher for patients with a secondary diagnosis of babesiosis [mean±SD, 9,451.47 ±11,608.34 vs. 4,003.57±1,459.74; $p = 0.073$]. It is important to note at this point that the cost

of care was similar for the 27 non-surgical cases but was higher for the 2 surgical cases [mean ±SD, 4,297.90±1,404.22 vs. 35,441.19±2,141.64; p<0.001].

## Discussion

Between January 1997 and December 2019 (23-year study period), a total of 29 cases were recorded in Spain (the incidence rate for the period was 0.28 cases per 10,000,000 person-years), which is almost triple the number of cases described in a similar previous study [28]. The distribution of cases was inconsistent, but it seemed to stabilize at 0.21 cases per 10,000,000 person-years in the last two years of the study. Thus, in Spain, babesiosis is still a rare infection with a high rate of asymptomatic patients [40]. These asymptomatic patients are capable of transmitting the disease, forcing us to maintain a high level of diagnostic suspicion in symptomatic patients (such as trips to endemic areas or transfusions in the last 6 months) [41].

In the United States, the number of cases reported to the Centers for Disease Control and Prevention practically doubled between 2011 and 2015, from 36.13 cases per 10,000,000 person-years to 64.63 cases per 10,000,000 person-years (i.e., from 1,126 to 2,074 cases) [42]. The highest annual incidence rate in Spain was 1.07 cases per 10,000,000 person-years in 2015 (5 cases described), which represents an incidence rate sixty times lower than that in the United States.

From the 29 patients, we registered 1 foreign case, 4 unknown cases and 24 autochthonous cases mainly distributed throughout the Castilla-La-Mancha and Extremadura regions, despite the literature describing a greater distribution of the *Ixodes* spp. tick in Castilla y León [43]. Only one other recent study describes the geographic distribution of *Ixodes* ticks across Spain, showing higher prevalences in Cantabria, Madrid and Andalucía [28]. Differences in incidence between autonomous regions could be explained by the existence of different biological cycles and the amount of cattle in the area, which determined the distribution of *Ixodes* ticks. However, it is curious that 85.19% (23) of our cases were of urban origin or perhaps this could simply be a limitation of the study. In our study, cases were clustered between June and September, without discrepancies with the previously described seasonality (May-September) [38, 39]. High temperature allows for faster egg laying, while high relative humidity allows longer survival of ovipositing females [44]. Three of our patients had Lyme disease, which is the most common coinfection described in the literature; this fact is not remarkable since *Babesia* spp. and *Borrelia burgdorferi sensu lato* share a common vector [43]. In addition, 4 cases (14.81%) were patients with HIV, a condition that has also been widely described since human babesiosis is more common in patients who are immunosuppressed, have hematological diseases, and those who have undergone a splenectomy, which are situations that increase parasitemia [45]. The male to female ratio of our patients was 5:1. This may be because the professions like farming, ranching, forestry among others that pose a higher risk of contracting babesiosis are mostly carried out by men. We have not found comparable data in the literature. The mean age of our patients was 45.6 years (±17.6), which is older than that described in a previous study carried out between 2004 and 2013 [28], and is perhaps more valuable given that our data encompasses a longer period of time. Of the 29 patients included in our study, 16 of them (55.2%) had a primary diagnosis of human babesiosis on their discharge report. These data, together with the fact that 23 of patients (79.3%) were hospital readmissions after a previous discharge, gives us a low clinical suspicion of human babesiosis.

Despite being a rare disease, babesiosis entails very high costs. The total hospital cost of inpatients diagnosed with human babesiosis in Spain from 1997 to 2019 was €186,925.66. (€6,445.71 mean cost per patient). Other infectious conditions requiring hospitalization in Spain, e.g., Q fever, total cost €154,232,779 (€36,600 mean cost per patient) [46];

strongyloidiasis, total cost €8,681,062.3 and mean cost per patient €17,122.4 [47]; the mean hospital admission cost was €5676, €104.2 million annually for all patients attended with a pneumococcal disease in Spanish hospitals [48]. Finally, the overall lethality rate of the cohort was 6.9%, with 2 deaths, both of which were cases of secondary diagnosis. This mortality rate is lower than that described in the literature (9% in general and up to 20% when transmission occurs by transfusion) [18, 49].

The death of these patients was not only due to *Babesia* infection, but this may have contributed to the fatal outcome. We aimed to relieve the lack of official epidemiological data, but we also hope to contribute to the development of hypotheses that will be worthy of exploration in further investigations.

The CMBD provides hospital data for more than 99% of the Spanish population. However, there is always a small percentage of lost information, so our data should be considered a very rough approximation of the epidemiological impact of babesiosis in Spain. The main limitations of our study are listed below. i) The likelihood of bias when collecting MBDS data is minimal compared to other health information systems, but the information included is not modifiable, so any coding error is irreversible; ii) the ICD-9 classification, used until 2015, includes fewer variables than those registered according to the ICD-10 classification, which was established later and provides more information; iii) the choice to include only patients hospitalized in public hospitals indicates a loss of information about those who were treated for the disease on an outpatient basis or requested medical assistance in primary care or in private hospitals, which means that hospital records underestimate the real burden of *Babesia* in Spain; iv) due to its low incidence, babesiosis is an uncommon disease, which leads to having low clinical suspicion in the majority of cases (asymptomatic or low/average clinical symptoms); v) not being able to access the medical history of patients prevented us from confirming the diagnosis and identifying the possible associated factors, such as a tick bite or transfusion, and does not provide information about tests used for *Babesia* diagnosis, which impairs the quality of the data; vi) given that only hospitalization costs have been included in the economic estimate, it is likely an underestimation since costs derived from, for example, sick leave have not been taken into account. In any case, our findings have potential implications for public policy. Therefore, as discussed above, these data underestimate the real incidence and the actual costs of *Babesia* in Spain during the period of this study.

In conclusion, we have demonstrated that human babesiosis is a rare zoonosis in Spain, but with an increasing incidence rate over the years, mainly in the regions of Castilla-La-Mancha and Extremadura. A low clinical suspicion for syndromes that are compatible with this infection is likely, so it is necessary to raise awareness among physicians to improve their detection. The mortality rate in our study was 6.9%, which was lower than that described in previous studies (9%). The inclusion of human babesiosis as a notifiable disease in endemic areas would be a useful tool for data collection to achieve a common national strategy to develop control measures, especially blood donor control.

## Supporting information

**S1 File.**
(SAV)

## Author Contributions

**Conceptualization:** Hugo Almeida, Javier Pardo-Lledías, Antonio Muro, Moncef Belhassen-García.

**Data curation:** Amparo López-Bernús.

**Formal analysis:** Montserrat Alonso-Sardón.

**Investigation:** Beatriz Rodríguez-Alonso, Ángela Romero-Alegría, Virginia Velasco-Tirado.

**Methodology:** Montserrat Alonso-Sardón.

**Supervision:** Antonio Muro, Moncef Belhassen-García.

**Writing – original draft:** Hugo Almeida, Moncef Belhassen-García.

**Writing – review & editing:** Amparo López-Bernús, Beatriz Rodríguez-Alonso, Montserrat Alonso-Sardón, Ángela Romero-Alegría, Virginia Velasco-Tirado, Javier Pardo-Lledías, Antonio Muro.

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
