## [Decision Letter · Decision Letter 0]

12 Sep 2022

PONE-D-22-19205IS BABESIOSIS A RARE ZOONOSIS IN SPAIN? ITS IMPACT ON THE SPANISH HEALTH SYSTEM OVER 23 YEARS.PLOS ONE

Dear Dr. Belhassen-García,

Thank you for submitting your manuscript to PLOS ONE. After careful consideration, we feel that it has merit but does not fully meet PLOS ONE’s publication criteria as it currently stands. Therefore, we invite you to submit a revised version of the manuscript that addresses the points raised during the review process. I apologize for the delay in submitting a recommendation.  Unfortunately, qualified reviewers are scarce and I must proceed with only one, and provide my own comments in lieu of a second reviewer.  The first reviewer made some very good comments and you must address those in your revision.

We look forward to receiving your revised manuscript.

Kind regards,

Sam R. Telford III

Academic Editor

PLOS ONE

Journal Requirements:

4. We note that Figures 1, 2 and 3 in your submission contain map images which may be copyrighted. All PLOS content is published under the Creative Commons Attribution License (CC BY 4.0), which means that the manuscript, images, and Supporting Information files will be freely available online, and any third party is permitted to access, download, copy, distribute, and use these materials in any way, even commercially, with proper attribution. For these reasons, we cannot publish previously copyrighted maps or satellite images created using proprietary data, such as Google software (Google Maps, Street View, and Earth). For more information, see our copyright guidelines: http://journals.plos.org/plosone/s/licenses-and-copyright.

 a. You may seek permission from the original copyright holder of Figures 1, 2 and 3 to publish the content specifically under the CC BY 4.0 license. 

Additional Editor Comments:

This is an interesting manuscript that summarizes an analysis of hospital billing code data representing human babesiosis cases in Spain. The data and analysis are a bit different than expectation, viz., that patients were mainly urban and that fatalities were in those secondarily admitted. The main infecting species in Spain has always been thought to be B. divergens (or B. venatarum, which is morphologically similar), in immune-compromised patients and with great morbidity and mortality. The ms would be greatly improved by a more thorough discussion of reference 32: what exactly did they find and how does the current work differ, as well as how the two studies agree. Some other points that need clarification:

1. What exactly is "missing data" for the exclusion criteria? Were there presumptive babesiosis cases that were not included in the analysis because of "missing data", and why?

2. There is no mention of splenectomy as a risk factor (comorbidity). This is a critical factor for analysis and should be included. Classically, B. divergens babesiosis is a medical emergency in splenectomized patients (e.g., Olmeda et al 1997 Acta Tropica, a case from the Canary Islands, technically part of Spain).

3. First paragraph of discussion: "high rate of asymptomatic patients". How do the authors know the asymptomatic to symptomatic ratio in Spain? It is possible that there are many asymptomatics but support for such a statement would require serosurveys, e.g., Pallas et al. 1999 doctoral thesis looking at risk in Galicia. (In that analysis, B. divergens seropositive people were from rural areas, with animal contact and reported tick bites).

4. The costs are emphasized. The figure by itself looks impressive, but is it really compared with other infectious conditions requiring hospitalization, e.g., pneumonia?

Reviewers' comments:

Reviewer's Responses to Questions

**Comments to the Author**

1. Is the manuscript technically sound, and do the data support the conclusions?

Reviewer #1: Yes

2. Has the statistical analysis been performed appropriately and rigorously? 

Reviewer #1: Yes

3. Have the authors made all data underlying the findings in their manuscript fully available?

Reviewer #1: Yes

4. Is the manuscript presented in an intelligible fashion and written in standard English?

Reviewer #1: Yes

5. Review Comments to the Author

Reviewer #1: In the Original Research Article: "IS BABESIOSIS A RARE ZOONOSIS IN SPAIN? ITS IMPACT ON THE SPANISH HEALTH SYSTEM OVER 23 YEARS." Almeida et al. analyzed data provided by hospital discharge records for patients with the diagnosis human babesiosis from 1997 to 2019 in Spain.

The authors found 29 patients diagnosed with babesiosis, 2 of them died. Data provided in the article contain information to kind of diagnosis (principal, primary, secondary), hospital stay, costs, co-morbidities, gender and age.

The study showed an increasing incidence rate over the years, most cases occurred in middle-aged men from urban areas between summer and autumn.

The presenting data are convincing to me for publication as such kinds of studies are rare, if even not existent in

Europe. Nonetheless, I recommend major revisions.

The attachment includes detailed comments.

6. PLOS authors have the option to publish the peer review history of their article (what does this mean?). If published, this will include your full peer review and any attached files.

Reviewer #1: No

---

## [Author Response · Author response to Decision Letter 0]

28 Sep 2022

Response to Editor

This is an interesting manuscript that summarizes an analysis of hospital billing code data representing human babesiosis cases in Spain. The data and analysis are a bit different than expectation, viz., that patients were mainly urban and that fatalities were in those secondarily admitted. The main infecting species in Spain has always been thought to be B. divergens (or B. venatarum, which is morphologically similar), in immune-compromised patients and with great morbidity and mortality. 

1. The ms would be greatly improved by a more thorough discussion of reference 32: what exactly did they find and how does the current work differ, as well as how the two studies agree. 

Thank you very much for your interesting comments. Study "32" (Guerrero et al.) is very similar to the one we present: a retrospective study of Babesia cases using the minimum basic dataset (CMBD in spanish) in the period 2004-2013, a total of 10 years (our study encompasses from 1997 to 2019, a period of 23 years). Our study period encompasses the study by Gerrero et al. in the 2 cut-off dates, which allows comparing their conclusions obtained between shorter dates, with ours, with a wider range of years and overlapping data between 2004 and 2013.

2. What exactly is "missing data" for the exclusion criteria? Were there presumptive babesiosis cases that were not included in the analysis because of "missing data", and why?

Missing data refers to insufficient data to corroborate the diagnosis of Babesiosis. All confirmed cases of Babesiosis were included in the study.

3. There is no mention of splenectomy as a risk factor (comorbidity). This is a critical factor for analysis and should be included. Classically, B. divergens babesiosis is a medical emergency in splenectomized patients (e.g., Olmeda et al 1997 Acta Tropica, a case from the Canary Islands, technically part of Spain).

Splenectomy is a risk factor for symptomatic babesiosis, but in the study we have not described the clinical component of admissions for babesiosis. We don’t have sufficient clinical data to describe that. 

4. First paragraph of discussion: "high rate of asymptomatic patients". How do the authors know the asymptomatic to symptomatic ratio in Spain? It is possible that there are many asymptomatics but support for such a statement would require serosurveys, e.g., Pallas et al. 1999 doctoral thesis looking at risk in Galicia. (In that analysis, B. divergens seropositive people were from rural areas, with animal contact and reported tick bites).

Thank you very much for your suggestion. We have included it.

5. The costs are emphasized. The figure by itself looks impressive, but is it really compared with other infectious conditions requiring hospitalization, e.g., pneumonia? 

Thank you very much for your suggestion. We have included it in the discussion.

Response to Reviewer

In the Original Research Article: "IS BABESIOSIS A RARE ZOONOSIS IN SPAIN? ITS IMPACT ON THE SPANISH HEALTH SYSTEM OVER 23 YEARS." Almeida et al. analyzed data provided by hospital discharge records for patients with the diagnosis human babesiosis from 1997 to 2019 in Spain. The authors found 29 patients diagnosed with babesiosis, 2 of them died. Data provided in the article contain information to kind of diagnosis (principal, primary, secondary), hospital stay, costs, co-morbidities, gender and age. The study showed an increasing incidence rate over the years, most cases occurred in middle-aged men from urban areas between summer and autumn.The presenting data are convincing to me for publication as such kinds of studies are rare, if even not existent in Europe. Nonetheless, I recommend major revisions.

6. Abstract: The authors write that 2.35% of hospitalized patients in Spain had the diagnosis babesiosis. I cannot imagine that this is right, because it would mean that if 29 patients are 2,35%, altogether only 1234 patients had been hospitalized in Spain (100%)? Please explain what does it mean:” The incidence of babesiosis in hospitalized patients in Spain is 2.35%.”

Thank you, we have corrected this mistake in the text. It refers to 2.35 cases/10,000,000 inhabitants/year.

7. Main findings and conclusions: The content is repeating, e.g. mortality rate, primary diagnosis, Please focus and shorten the abstract. 

We have modificate and shorten the abstract.

8. Last sentence: “ 100% of fatal cases with secondary diagnosis” – I think you should not write 100% because the number of patients who died is only 2. Better: “Both patients who died,…”

Thank you, we have corrected this mistake in the text.

9. Introduction: The introduction is too long. Please shortened it, e.g. details on ticks, clinic of the disease, There are god reviews you can refer to, e.g. (Hildebrandt et al. 2021)

Thank you very much for your suggestion. We shortened the introduction and we have included the suggested review.

10. The therapy is not only atovaquone + azithromycin, but also clindamycin + quinine, dependant on the species and severity of the disease.(Smith et al. 2020)

Thank you very much for your suggestion. We have included it.

11. The authors write that since 2003 only 4 diagnosed cases have been published in Spain: 3x B. divergens, 1x B. microti. Accorging to the review from Hildebrandt et al., 2021(Hildebrandt et al. 2021) there were at least 5 cases of B. microti (Arsuaga et al. 2016, de Ramon et al. 2016, Merino 2016, Arsuaga et al. 2018, Guirao-Arrabal et al. 2020) and 3 cases of B. divergens (Gonzalez et al. 2014, Gonzalez et al. 2015, Asensi et al. 2018) published from Spain since 2003. Please add these references.

Many thanks for the reviewer comment, we added the number of the total reported cases in Spain with the mentioned references.

12. Material and methods: Data collection: The passage about the ICD-9 and ICD-10 is too detailed. Please shorten this passage.

We have shorten ICD-9 and ICD-10 description.

Results: Clinical data. 

13. The authors write that 23 of the 29 cases were hospital readmissions. What does this mean? Why were these patients readmitted – because of babesiosis or because of another disease? What were the diagnoses these patients were admitted before and what time (days, weeks) counted as readmission?

Readmission is defined as any admission of a patient within 30 days of discharge (as cited in the manuscript). The reasons for readmission cannot be known due to the methodology of the study.

Discussion

14. What are the reason that the cited previous study (32) found a lower incidence of babesiosis in Spain, especially with the information that between 2017 and 2019 only 3 additional cases were reported? There are more than 1 imported cases, see comments above.

Thank you very much for your interesting comments. Study "32" (Guerrero et al.) is very similar to the one we present: a retrospective study of Babesia cases using the minimum basic dataset (CMBD in spanish) in the period 2004-2013, a total of 10 years (our study encompasses from 1997 to 2019, a period of 23 years). Our study period encompasses the study by Gerrero et al. in the 2 cut-off dates, which allows comparing their conclusions obtained between shorter dates, with ours, with a wider range of years and overlapping data between 2004 and 2013.

15. How many of the 23 urban cases received blood transfusion, because you write that accessibility of blood transfusion could be a reason for higher incidence in urban areas?

It was a typing error at the time of translation. Thank you, we have corrected this mistake in the text.

16. 23 patients were readmissions: see comment above: readmissions because of babesiosis or because of another disease? What were the diagnoses these patients were admitted before and what time (days, weeks) counted as readmission? – does it only mean that these patients with readmissions had co-morbidities, e.g. because of age,.. 

Readmission is defined as any admission of a patient within 30 days of discharge (as cited in the manuscript). The reasons for readmission cannot be known due to the methodology of the study.

17. The reported mortality varies according to the species. What species were identified in the 29 patients? – How many patients had species identifications. What were the methods for diagnosis and what therapy got the two patients who died?

Unfortunately, we cannot obtain these data due to the methodology of the study.

Journal Requirements

18. Your ethics statement should only appear in the Methods section of your manuscript. If your ethics statement is written in any section besides the Methods, please delete it from any other section.

Thank you, we have corrected this mistake in the manuscript.

19. We note that Figures 1, 2 and 3 in your submission contain map images which may be copyrighted. All PLOS content is published under the Creative Commons Attribution License (CC BY 4.0), which means that the manuscript, images, and Supporting Information files will be freely available online, and any third party is permitted to access, download, copy, distribute, and use these materials in any way, even commercially, with proper attribution. 

Figure 3 is not copyrighted despite its use in other studies. Figures 1 and 2 are graphs.

Yours sincerely, Moncef Belhassen García

---

## [Decision Letter · Decision Letter 1]

12 Dec 2022

PONE-D-22-19205R1Is Babesiosis a rare zoonosis in Spain? Its impact on the Spanish Health System over 23 years.PLOS ONE

Dear Dr. Belhassen-García,

Thank you for submitting your manuscript to PLOS ONE. After careful consideration, we feel that it has merit but does not fully meet PLOS ONE’s publication criteria as it currently stands. Therefore, we invite you to submit a revised version of the manuscript that addresses the points raised during the review process.

ACADEMIC EDITOR: 

The article has been greatly improved but needs minor revisions before being published.

We look forward to receiving your revised manuscript.

Kind regards,

Maria Stefania Latrofa

Academic Editor

PLOS ONE

Journal Requirements:

Additional Editor Comments:

I would suggest deleting figure 2 and briefly describing the data (infection prevalence in relation to the month) in the text.

Reviewers' comments:

Reviewer's Responses to Questions

**Comments to the Author**

1. If the authors have adequately addressed your comments raised in a previous round of review and you feel that this manuscript is now acceptable for publication, you may indicate that here to bypass the “Comments to the Author” section, enter your conflict of interest statement in the “Confidential to Editor” section, and submit your "Accept" recommendation.

Reviewer #2: (No Response)

2. Is the manuscript technically sound, and do the data support the conclusions?

Reviewer #2: Yes

3. Has the statistical analysis been performed appropriately and rigorously? 

Reviewer #2: N/A

4. Have the authors made all data underlying the findings in their manuscript fully available?

Reviewer #2: Yes

5. Is the manuscript presented in an intelligible fashion and written in standard English?

Reviewer #2: Yes

6. Review Comments to the Author

Reviewer #2: The manuscript "PONE-D-22-19205R1" analyses the cases of human Babesiosis reported in Spain from 1997 to 2019. Authors analyzed 29 cases, with two deaths, and also reported that most cases were from men patients. In my opinion, the data is worth to be published, however, there are some points in the current version that needs to be revised by authors before it becomes suitable for publication in PLOS ONE.

In the conslusions section of the abstract authors wrote "Given the low rate of primary diagnoses (55.2%) and the high number of readmissions (79.3%), a low clinical suspicion is likely. There was a 6.9% mortality in our study. Both patients who died were patients with secondary diagnoses of the disease." In my view this should be maintained in the main findings section as results should not appear for the first time in the conclusions.

In the first and second lines of the introduction I suggest to change "...and transmitted by" by ", which is mainly transmitted by"

I suggest to delete the sentence "Ticks are parasites found worldwide that have great epidemiological and clinical importance; especially in terms of human health (3,4)and veterinary medicine, affecting 80% of the world´s cattle population (5)." as it does not add any relevant information, and seems to be a lost sentence as ticks are not the main topic of the manuscript.

In the last line of the first paragraph of introduction replace "Ixodes species" by "tick species"

In first lines of the second paragraph of introduction, please specify in which hosts these prevalence were recorded.

In the third paragraph of discussion, please add "sensu lato" after "Borrelia burgdorferi".

Still in the third paragraph of introduction the authors wrote "Of the 29 patients included in our study, only 16 of them (55.2%) had a primary diagnosis of human babesiosis on their discharge report." I'm not sure that it is a low number, more than half of the patients had babesiosis as primary diagnosis. Does the authors checked if this number should be considered low. Is there data for other diseases that may explain why it should be considered low? In addition, the sentence "leads us to think of a very low clinical suspicion is likely human babesiosis" is not clear and shoud be rephrased.

Please delete "We have also to declare that" from the fifth paragraph and start the sentence with "The death of these patients..."

7. PLOS authors have the option to publish the peer review history of their article (what does this mean?). If published, this will include your full peer review and any attached files.

Reviewer #2: No

---

## [Author Response · Author response to Decision Letter 1]

19 Dec 2022

Response to Editor

1. I would suggest deleting figure 2 and briefly describing the data (infection prevalence in relation to the month) in the text.

Thank you very much for your suggestion. We have deleted figure 2 and included a brief description in the text.

Response to Reviewer

The manuscript "PONE-D-22-19205R1" analyses the cases of human Babesiosis reported in Spain from 1997 to 2019. Authors analyzed 29 cases, with two deaths, and also reported that most cases were from men patients. In my opinion, the data is worth to be published, however, there are some points in the current version that needs to be revised by authors before it becomes suitable for publication in PLOS ONE.

Abstract

2. In the conclusions section of the abstract authors wrote "Given the low rate of primary diagnoses (55.2%) and the high number of readmissions (79.3%), a low clinical suspicion is likely. There was a 6.9% mortality in our study. Both patients who died were patients with secondary diagnoses of the disease." In my view this should be maintained in the main findings section as results should not appear for the first time in the conclusions.

Thank you very much for your suggestion. We have changed it: we also included the data in the main fingings section.

Introduction

3. In the first and second lines of the introduction I suggest to change "...and transmitted by" by ", which is mainly transmitted by"

Thank you very much for your suggestion. We have changed it.

4. I suggest to delete the sentence "Ticks are parasites found worldwide that have great epidemiological and clinical importance; especially in terms of human health (3,4)and veterinary medicine, affecting 80% of the world´s cattle population (5)." as it does not add any relevant information, and seems to be a lost sentence as ticks are not the main topic of the manuscript.

Thank you, we have deleted this sentence. 

5. In the last line of the first paragraph of introduction replace "Ixodes species" by "tick species"

Thank you very much for your suggestion. We have changed it.

6. In first lines of the second paragraph of introduction, please specify in which hosts these prevalence were recorded.

Thank you very much for your suggestion. We have included it. The main hosts were cattle roe deer and small mammals, like mice.

Discussion

7. In the third paragraph of discussion, please add "sensu lato" after "Borrelia burgdorferi".

Thank you very much for your suggestion. We have included it.

8. Still in the third paragraph of discusion the authors wrote "Of the 29 patients included in our study, only 16 of them (55.2%) had a primary diagnosis of human babesiosis on their discharge report." I'm not sure that it is a low number, more than half of the patients had babesiosis as primary diagnosis. Does the authors checked if this number should be considered low. Is there data for other diseases that may explain why it should be considered low? In addition, the sentence "leads us to think of a very low clinical suspicion is likely human babesiosis" is not clear and shoud be rephrased.

Thank you very much for your suggestion. We did not check whether this number should be considered low. We made the changes in the text.

9. Please delete "We have also to declare that" from the fifth paragraph and start the sentence with "The death of these patients..."

Thank you, we have corrected this mistake in the text. 

Yours sincerely, Moncef Belhassen Garcia

---

## [Editor Report · Decision Letter 2]

21 Dec 2022

Is Babesiosis a rare zoonosis in Spain? Its impact on the Spanish Health System over 23 years.

PONE-D-22-19205R2

Dear Dr. Belhassen-García,

We’re pleased to inform you that your manuscript has been judged scientifically suitable for publication and will be formally accepted for publication once it meets all outstanding technical requirements.

Kind regards,

Maria Stefania Latrofa

Academic Editor

PLOS ONE

---

## [Editor Report · Acceptance letter]

24 Jan 2023

PONE-D-22-19205R2 

Is Babesiosis a rare zoonosis in Spain? Its impact on the Spanish Health System over 23 years. 

Dear Dr. Belhassen-García:

I'm pleased to inform you that your manuscript has been deemed suitable for publication in PLOS ONE. Congratulations! Your manuscript is now with our production department. 

Kind regards, 

on behalf of

Dr. Maria Stefania Latrofa 

Academic Editor

PLOS ONE